# Role of TRP Channels in Liver-Related Diseases

**DOI:** 10.3390/ijms241512509

**Published:** 2023-08-07

**Authors:** Yusheng Liu, Yihan Lyu, Lijuan Zhu, Hongmei Wang

**Affiliations:** 1Department of Pharmacology, School of Medicine, Southeast University, Nanjing 210009, China; 213191092@seu.edu.cn (Y.L.); 213191078@seu.edu.cn (Y.L.); 2Key Laboratory of Developmental Genes and Human Diseases, MOE, Department of Histology and Embryology, School of Medicine, Southeast University, Nanjing 210009, China; zhulj@seu.edu.cn

**Keywords:** TRP channels, liver diseases, liver injury, liver fibrosis, hepatocellular carcinoma

## Abstract

The liver plays a crucial role in preserving the homeostasis of an entire organism by metabolizing both endogenous and exogenous substances, a process that relies on the harmonious interactions of hepatocytes, hepatic stellate cells (HSCs), Kupffer cells (KCs), and vascular endothelial cells (ECs). The disruption of the liver’s normal structure and function by diverse pathogenic factors imposes a significant healthcare burden. At present, most of the treatments for liver disease are palliative in nature, rather than curative or restorative. Transient receptor potential (TRP) channels, which are extensively expressed in the liver, play a crucial role in regulating intracellular cation concentration and serve as the origin or intermediary stage of certain signaling pathways that contribute to liver diseases. This review provides an overview of recent developments in liver disease research, as well as an examination of the expression and function of TRP channels in various liver cell types. Furthermore, we elucidate the molecular mechanism by which TRP channels mediate liver injury, liver fibrosis, and hepatocellular carcinoma (HCC). Ultimately, the present discourse delves into the current state of research and extant issues pertaining to the targeting of TRP channels in the treatment of liver diseases and other ailments. Despite the numerous obstacles encountered, TRP channels persist as an extremely important target for forthcoming clinical interventions aimed at treating liver diseases.

## 1. Introduction

The liver serves as a vital organ with a primary metabolic function. In addition, it possesses the ability to synthesize and decompose proteins, regulate overall blood volume, eliminate toxins, and regulate immunity, all of which are crucial for maintaining normal physiological activities in the human body [1]. Lately, there has been a steady rise in the number of individuals diagnosed with liver diseases globally, thereby causing a significant public health concern. Liver disease is a multifaceted and progressive pathological condition that results from various internal and external pathogenic factors, such as viral infections, chemical exposure, drug use, malnutrition, and acid–base imbalances [2,3,4]. Chronic viral hepatitis, alcohol-related liver disease, and nonalcoholic fatty liver disease (NAFLD) are all potential causes of liver fibrosis, which can ultimately lead to the development of cirrhosis and liver cancer [5,6,7]. Based on epidemiological data, the global annual mortality rate resulting from end-stage liver diseases, including hepatitis, cirrhosis, and hepatocellular carcinoma (HCC), can reach up to 2 million individuals [8]. Despite this high mortality rate, there exists a dearth of treatment options. Consequently, the creation of targeted therapeutic medications for liver diseases holds immense importance for clinical intervention.

The transient receptor potential superfamily comprises non-selective cation channels, with coding proteins that exhibit widespread distribution in mammalian cells, including humans. TRP channels are categorized based on variations in amino acid sequence and topological structures. They are further divided into seven families: TRP canonical (TRPC), TRP vanilloid (TRPV), TRP melastatin (TRPM), TRP ankyrin (TRPA), TRP polycystic (TRPP), TRP mucolipin (TRPM), and TRP no mechanoreceptor potential C (NOMPC) (TRPN) [9,10,11]. TRP channels have a broad distribution across various organs, including the heart, kidneys, testes, lungs, liver, spleen, ovaries, intestine, prostate, placenta, uterus, and vascular tissues, as reported in the literature [12,13]. These channels are known to serve multiple functions in sensory perception, vasodilation, and cell proliferation, primarily by facilitating the influx of monovalent and divalent cations (e.g., Na^+^, Mg^2+^, and Ca^2+^) and trace metal ions. Moreover, TRPs facilitate the modulation of apoptosis activation or inhibition and cell migration proficiency by regulating intracellular ions, thereby mediating the physiological and pathophysiological processes of numerous cancer and immune system cells [14,15,16]. Given that TRPs are predominantly situated on the cellular surface, they have been investigated as potential targets for analgesics and other pharmaceuticals. Many compounds targeting TRPV1, TRPV4, TRPA1, and TRPM8 channels have been included in previous clinical trials [17,18].

Numerous members of the TRP channel family have been found to be expressed in the liver cells, including isolated hepatocytes and immortalized hepatocyte lines. These channels play a crucial role in regulating the homeostasis of intracellular cation levels and various biological functions of the liver, such as the regulation of glucose, fatty acids, amino acids, exogenous metabolism, bile acid secretion, protein synthesis, and secretion [19]. Therefore, the modulation of TRP activity is expected to exert an impact on the incidence and progression of hepatic disorders. In this study, we comprehensively appraised the physiological roles of TRP channels in diverse cell types in the hepatic tissue. Furthermore, we discussed the involvement of established TRP channels in liver pathologies, including acute and chronic hepatic injury, hepatitis, liver fibrosis, and liver carcinoma, with the aim of enhancing our comprehension of the significance of TRP channels in hepatic diseases and facilitating the identification of prospective therapeutic targets. First, we briefly summarize the progress in the treatment of liver fibrosis and HCC to establish the current situation that liver diseases are difficult to reverse using drug therapy and further determine the significance of finding more drug targets.

## 2. Advances in the Treatment of Liver Diseases

The progression of chronic inflammatory diseases is often characterized by organ fibrosis. Chronic liver inflammation and fibrosis are primarily caused by alcohol-related liver disease, viral hepatitis, and nonalcoholic fatty liver disease (NAFLD), which are also major contributors to the risk of HCC. Liver cirrhosis, a consequence of extensive liver fibrosis, is currently the 11th leading cause of death worldwide, resulting in approximately 2 million deaths annually. Therefore, the development of effective anti-fibrosis drugs is of utmost importance [20,21]. According to research, cirrhosis is no longer deemed to be an irreversible and progressive disease after the elimination of its etiology [22,23]. In recent years, a range of in vitro and in vivo models have been established by researchers to facilitate the development of anti-fibrosis drugs [24,25,26].

Selonsertib, a selective apoptosis signal-regulating kinase (ASK1) inhibitor, has demonstrated potential in ameliorating the fibrosis process in the nonalcoholic steatohepatitis (NASH) mouse model. Furthermore, NASH F2-3 patients who underwent a 24-week treatment exhibited an improvement in their histological fibrosis degree [27]. The activation of hepatic stellate cells (HSCs) is facilitated by NOX 1, 2, and 4, which generate superoxide free radicals, thereby contributing to the progression of liver fibrosis. GKT137831, a dual inhibitor of NOX1/4, has been found to significantly reduce CCl4-induced liver fibrosis in mice by inhibiting the production of reactive oxygen species (ROS) in HSCs both in vitro and in vivo. Regression of histological fibrosis in metabolic liver disease can be induced by lifestyle changes and weight loss surgery [28]. Moreover, the removal of chronic inflammation has been demonstrated as the primary inducer for the regression of advanced liver fibrosis caused by chronic HBV and HCV infections [29,30]. Several plant-derived medications have been found to exhibit a diverse range of protective effects against liver fibrosis [31,32]. In patients with mild hepatic encephalopathy and cirrhosis, the administration of four probiotics (*Clostridium* valerate, *Bacillus mesentericus*, *Lactobacillus paracasei*, and *Lactiplantibacillus plantarum*) and three fibers (β-glucan, pectin, and resistant starch) has been associated with a reduction in serum TBIL levels and an increase in albumin levels. Despite these findings, there is currently no FDA-approved anti-fibrosis medication available for clinical use. The only available treatment option for patients with advanced cirrhosis is liver transplantation [33].

HCC is a prevalent type of liver cancer and is the third leading cause of cancer-related death. Despite the gradual increase in its incidence, treatment options remain limited. Immune checkpoint inhibitor (ICI) therapy has become an essential component of systemic treatment for advanced HCC [34]. Studies have demonstrated that a combination of epidermal growth factor receptor (EGFR) inhibitor gefitinib and levatinib exerts a potent anti-proliferative effect on EGFR-expressing liver cancer cell lines in vitro, as well as HCC tumors in vivo [35]. Erlotinib, an EGFR inhibitor, either alone or in combination with sorafenib, has demonstrated certain advantages for patients with HCC [36]. Advanced HCC that is fgf19-positive is driven by FGFR4. Fisogatinib, a highly selective FGFR4 inhibitor, has been shown to be both tolerable and effective in treating advanced HCC with FGF19 expression, thus clinically validating the therapeutic potential of targeting the FGF19-FGFR4 pathway [37]. Furthermore, the combination of LXRα agonists and RAF inhibitors or CDC7 inhibitors and mTOR inhibitors may further enhance the therapeutic approach toward HCC [38,39]. However, there is only one small molecular drug Sorafenib (Nexavar^®^) that is approved by the FDA for the treatment of advanced HCC [40]. Therefore, more effective potential therapeutic targets still need to be explored.

The modulation of ion channel expression has the potential to govern cellular functions such as survival, proliferation, differentiation, apoptosis, and tumorigenesis [41,42]. In contrast to other ion channel families, TRP channels exhibit distinct activation mechanisms and roles. Studies have demonstrated the expression of numerous TRP channel subtypes within the liver [43]. The influx of cations, facilitated by TRP channels, exhibits a close association with various cellular processes such as proliferation, differentiation, and apoptosis of hepatocytes and HCC cells, generation of vascular endothelial cells, activation of HSCs, and activation and migration of KCs. In the context of liver inflammation, TRP channels augment the inflammatory response following hepatocyte injury by regulating biological functions such as cytokine and chemokine production, chemokine response, and adhesion and migration of KCs, monocytes, and neutrophils [44]. The TRPV1, TRPC1, TRPM2, and TRPM7 channels serve as the foundation for Ca^2+^ entry and subsequent cellular damage induced by paracetamol in human hepatoma (HepG2) cells [45]. Further, TRPM8 activation by environmental stimuli plays a critical role in numerous cancers and inflammation-related diseases. The deletion of TRPM8 attenuates liver fibrosis through the S100A9-HNF4α signaling pathway [46]. In HepG2 and Huh-7 cells, TRPC6 and plasma membrane (PM) (Na^+^-Ca^2+^) exchanger (NCX1) are essential components of the cytokine transforming growth factor-beta (TGF-β), which promotes HCC tumorigenesis and progression [47]. The TRP channel plays a crucial role in liver-related pathologies, with ongoing investigations into the expression and function of TRP channels in various liver cell types. The targeting of TRP channels represents a potential alternative strategy for the clinical management of liver diseases in the future.

## 3. Expression and Function of TRP Channels in Different Types of Cells in the Liver Tissue

The liver has a pivotal function in preserving systemic homeostasis through the concerted efforts of various cell types, including hepatocytes, sinusoidal endothelial cells, KCs, smooth muscle cells, HSCs, and oval cells. The hepatocytes, which are hexagonal lobules, are separated from the sinusoidal endothelial cells by a Disse gap, where HSCs are situated. By releasing cytokines and extracellular matrix (ECM) components, KCs, which are resident macrophages derived from monocytes that reside in the sinus cavity, maintain sinusoidal tone and liver hardness, serving as the first line of defense of the liver immune system. Further, various types of liver cells express numerous members of the TRP protein family (Table 1).

### 3.1. TRP Channels in Hepatocytes

Hepatocytes (parenchymal cells) are the main cell type in the liver and account for approximately 70% of all liver cells [78]. Consistent with the complex function and structure of the liver, hepatocytes are highly differentiated cells with spatial polarization and characteristic intracellular signaling. The modulation of calcium concentration within hepatocytes is a crucial intracellular signaling network that governs the physiological and pathological function of both the hepatocytes and the liver. The alteration of cytoplasmic free calcium concentration ([Ca^2+^]_cyt_) within hepatocytes regulates various metabolic processes such as glucose, fatty acid, amino acid, and xenobiotic metabolism; bile acid secretion; protein synthesis and secretion; lysosome and other vesicular movement; and cell proliferation, apoptosis, and necrosis [79,80,81,82]. Furthermore, the modulation of calcium ion levels within distinct organelles of hepatocytes is a crucial factor in regulation. In particular, the concentration of calcium ions within the mitochondria ([Ca^2+^]_mt_) governs the citric acid cycle, ATP synthesis, and cell apoptosis [83], whereas the concentration of calcium ions within the endoplasmic reticulum (ER) regulates protein synthesis and the metabolism of xenobiotic compounds. In addition, the concentration of calcium ions within the nucleus is involved in the regulation of cell proliferation [84]. The intracellular Ca^2+^ signaling pathway mainly includes intracellular storage of Ca^2+^ in the ER and mitochondria, the release of Ca^2+^ from ER to the cytoplasm induced by IP3, and Ca^2+^ entry into the cell through the PM mediated by PM and ER (Ca^2+^ + Mg^2+^) ATP-ase [78,81]. The elevation of [Ca^2+^]_cyt_ is primarily attributed to augmented Ca^2+^ discharge from the ER and mitochondria, and/or Ca^2+^ influx through the PM. The Ca^2+^ permeable channel situated on the PM is regulated by storage-operated Ca^2+^ (SOC) channels, receptor activation, stretch activation, and ligand gating. Notably, the TRPs are Ca^2+^ permeable channels predominantly mediated by receptor activation in hepatocytes [73,85].

Ca^2+^ influx mediated by TRPV1 activation is involved in human hepatoma HepG2 cell migration via a number of calcium-sensitive targets such as myosin light chain kinase, gel protein, non-actin, and calcineurin. HGF may facilitate the activation of TRPV1 channels in the PM or augment the recruitment of expressed channels to the PM. Furthermore, the findings suggest that TRPV1 activation reduces the interaction between the C-terminal region of TRPV1 and microtubules, thereby compromising the stability of microtubules [86]. Waning et al. postulated that the activation of TRPV1, situated at the leading edge of HepG2 cells, by low-dose capsaicin, resulted in the augmentation of microtubule asymmetry. This, in turn, altered the dynamics of microtubules at the rear end of the cells, rendering them highly susceptible to shrinkage and, consequently, promoting cell migration. Nonetheless, it is imperative to conduct further investigations to ascertain the surface distribution of endogenous TRPV1 channels in HepG2 cells and to determine the feasibility of co-localization with microtubules [48]. Furthermore, it has been established that 4α-Phosphate-12,13-disdecanoate (4α-PDD) and arachidonic acid are activators of TRPV4, and elicit Ca^2+^ influx in HepG2 cells, thereby indicating the presence of functional TRPV4 in this particular hepatocyte lineage [58]. TRPV4 plays a crucial role in various physiological processes by detecting osmotic pressure, pressure, heat, and ROS and also serves as a significant determinant of acetaminophen (APAP)-induced hepatotoxicity [87,88].

The TRPM2 channel functions as a non-selective cation channel that is permeable to Ca^2+^. The binding of intracellular ADP ribose (ADPR) and the TRPM2 NUDT9-h motif at the C-terminal facilitates the opening of channel pores [89,90,91]. Immunofluorescence and confocal microscopy studies have shown that the functional TRPM2 channel is predominantly located in organelles within rat hepatocytes [71]. Moreover, the activation of TRPM2-mediated Ca^2+^ influx has been linked to the induction of apoptosis and necrosis pathways in various cell types, ultimately resulting in cell death. Further, there is evidence suggesting that TRPM2 is implicated in oxidative stress-mediated hepatocyte injury. Mishra et al. discovered that the activity of TRPM7 in resting rat hepatocytes is inferior to that in proliferating WIF-B cells, a space-polarized cell line derived from a hybrid of rat liver cancer and human skin fibroblasts. This finding implies that TRPM7 may be associated with hepatocyte proliferation [92]. In rat H4-IIE hepatocytes, TRPM7 is responsive to ROS-mediated Ca^2+^, Na^+^, and Mg^2+^ influx as well as to variations in intracellular ATP concentration, which can result in apoptosis or necrosis [73].

The upregulation of TRPC1 in the H4-IIE rat hepatoma cell line has been reported to result in notable augmentation of Ca^2+^ and Na^+^ influx in response to maintoxin. Conversely, the downregulation of TRPC1 led to a reduction in maintoxin-induced Ca^2+^ entry, an increase in swelling, and a decrease in regulatory volume in a hypotonic solution, indicating that TRPC1 plays a crucial role in regulating hepatocyte volume [61,93]. Furthermore, research has demonstrated that TRPC1 and TRPV1 are responsible for the Ca^2+^ influx and subsequent cellular damage caused by APAP in HepG2 cells [45]. TRPML1, another TRP channel, functions as a non-selective cation channel that facilitates the influx of Ca^2+^, Na^+^, and K^+^. Its expression has been confirmed in lysosomes of rat hepatocytes, where it plays a crucial role in regulating intracellular Ca^2+^ homeostasis by facilitating the release of Ca^2+^ from lysosomes [77].

### 3.2. TRP Channels in HSCs

The HSCs are nonparenchymal cells found in the liver, which are also referred to as fat storage cells, Ito cells, fat cells, interstitial cells, perivascular cells, or vitamin A storage cells. The HSCs are characterized by an irregular astrocyte body with a round or oval nucleus, and the lipid droplets in these cells store palmitic acid retinol containing vitamin A [94,95]. Under normal conditions, HSCs remain in a non-proliferative static state, with minimal expression of α-smooth muscle actin (α-SMA) and low collagen synthesis ability [96]. These cells are typically located in Disse space and exhibit attributes akin to resident fibroblasts, being embedded within the normal matrix, as well as pericytes, which are endothelial cells that attach to capillaries [95,97,98]. However, upon activation, HSCs undergo trans-differentiation into myofibroblasts, which possess both proliferative and contractile properties and are responsible for generating a significant amount of ECM deposits. This process represents a critical step in the progression of liver fibrosis following liver injury [99,100].

TRPM7, a non-selective cation channel, was initially identified and cloned in 2001 with PCR using mouse brain cDNA. This channel exhibits a notable permeability to Ca^2+^, Mg^2+^, and Na^+^ [101,102]. Its involvement in a diverse range of physiological and pathological processes, such as cell proliferation, survival, migration, adhesion, embryonic development, hypoxia/ischemia-induced neuronal death, apoptosis, and tumor cell metastasis, has been established using research. Furthermore, TRPM7 is expressed in human, mouse, and zebrafish liver tissues, as well as various hepatocytes including HSCs, suggesting its potential role in the survival of HSCs [92,103]. Further, the activation of caspase-3 by TNF-related apoptosis inducing-ligand (TRAIL) results in the induction of apoptosis in activated HSC-T6 cells. Inhibition of TRPM7 with Gd^3+^ or 2-aminoethoxydiphenyl borate (2-APB) reduced the expression of α-SMA and type I collagen (Col1α1) in HSC-T6 cells that were activated following TRAIL pretreatment, indicating the involvement of TRPM7 in TRAIL-induced HSC apoptosis [76,104]. Following TRPM7 blockage, an elevation in beclin-1 levels and a reduction in bmf mRNA levels were observed in HSC-T6 cells, which is potentially linked to the activation of Bax and Bcl-2 factors [104]. Furthermore, the phosphorylated forms of extracellular signal-regulated kinase (ERK) and protein kinase B (AKT) in HSC-T6 cells were found to decrease, suggesting that ERK and AKT serve as downstream signals of TRPM7 [75]. Similarly, PI3K and ERK inhibition has been reported to impede the expression of type I collagen and α-SMA, as well as cell proliferation. In conclusion, stimulators mediate HSC activation by activating the ERK and PI3K/AKT pathways, and TRMP7-mediated cation influx helps with the continued activation of the PI3K and ERK pathways in response to cell proliferation [105,106].

The TRPV4 channel, a subclass of the TRP superfamily channels, exhibits abnormal expression in liver fibrosis [107]. The multiple activation and regulation sites of this channel enable it to integrate diverse environmental stimuli and mediate various physiological functions, including cell proliferation, survival, differentiation, migration, and adhesion. Further, TGF-β1, the most potent fibroblast factor in the liver, induces HSC activation and significantly increases TRPV4 expression in rat HSC-T6 cells [108,109]. The administration of a selective agonist, 4α-PDD, to TRPV4 significantly suppresses the expression of pro-caspase3 protein and Bax, both of which are associated with apoptosis, and subsequently stimulate autophagy in HSC-T6. Conversely, the inhibition of TRPV4 expression using specific TRPV4 siRNAs significantly increases the expression of caspase3 mRNA, pro-caspase3 protein, and Bax, leading to the inhibition of HSC-T6 autophagy [59]. These findings suggest that TRPV4 plays a crucial role in HSC activation and proliferation. The transient receptor potential vanilloid 3 (TRPV3) exhibits broad distribution in skin keratinocytes, oral and nasal epithelium, and the nervous system, liver, and kidneys and possesses high Ca^2+^ permeability. Its association with dermal fibroblasts has been established [110,111]. Yan et al. conducted an immunohistochemical analysis and observed a significant elevation in TRPV3 levels in cirrhotic liver tissue compared with normal liver tissue. In HSCs, the inhibition of DNA synthesis and the promotion of cell apoptosis can be achieved with the knockdown of TRPV3 using siRNA. Furthermore, the downregulation of TRPV3 has been shown to decrease the levels of lectin-like oxidized low-density lipoprotein receptor-1 (LOX-1) protein. In vivo investigations have demonstrated a positive correlation between LOX-1 expression and the level of inflammatory factors [56,112]. Consequently, a reduction in TRPV3 expression or function can impede the proliferation of HSCs and enhance an inflammatory response [57]. In addition, the inhibition of TRPV3 can decrease the proliferation of cardiac fibroblasts through the TGF-β1/CDK2/cyclin E pathway [113]. This indicates that the TRPV3 channel plays a role in the process of fibroblast and tissue fibrosis.

### 3.3. TRP Channels in KCs

KCs are nonparenchymal liver cells that function as tissue macrophages within liver sinuses, playing a crucial role in the regulation of the liver immune system. KCs contribute to the immune response of the liver by engaging in phagocytosis, presenting antigens, and producing proinflammatory cytokines and pro-fibrosis factors, thereby instigating inflammation, necrosis, regeneration, and fibrosis [114,115].

Evidence suggests that the expression of TRP channels is associated with the mediation of liver inflammation activated by KCs. However, there is a lack of literature on the specific expression of TRP channels in KCs. Notably, in TRPV1-knockout mice, the levels of chemokines (Ccl2 and Cxcl2) and proinflammatory cytokines (Tnf-a, Il-1a, Il-1b, and Il-6) in the liver were significantly reduced, along with a decrease in neutrophil infiltration [49]. The activation of KCs resulted in the generation of superoxide and proinflammatory cytokines, which in turn facilitated the recruitment of monocytes and neutrophils, culminating in hepatic inflammation. The proposition posits that TRPs have the potential to augment an inflammatory response by modulating the biological processes of KCs, such as inducing cytokine and chemokine production. The activation of KCs is attributed to calcium overload, which is a significant factor. Jiang et al. used the liver ischemia/reperfusion (I/R) injury model to observe the escalation of store-operated calcium channel currents and the activation of phospholipase C in KCs [116]. In intact cells, receptor-mediated activation of phospholipase C activates the production of TRPC3 diacylglycerol independent of G protein, protein kinase C, and inositol 1,4,5-triphosphate. Hepatic ischemia/reperfusion (I/R) injury has been identified as a potential mediator of Ca^2+^ influx through the activation of SOC channels or TRPC3 channels, resulting in calcium overload and subsequent activation of KCs. Elaidic acid (EA) has been found to induce NLRP3 inflammasome activation in KCs through the activation of the MAPK signaling pathway, which is mediated by the endoplasmic reticulum stress (ERS) response. This activation ultimately leads to the release of IL-1β and IL-18, resulting in an inflammatory response [63]. Although the downstream signaling pathway of ERS has been elucidated, the mechanism by which electroacupuncture (EA) mediates the ERS response remains uncertain. Calcium, functioning as a second messenger, modulates the activity of various regulatory proteins, including chaperones, enzymes, and transcription factors, thereby influencing ERS and autophagy [117,118,119]. TRP channels function as nonselective cation channels, and their expression and activity play a critical role in regulating intracellular calcium concentration, thereby modulating all Ca^2+^-dependent processes. Hence, it is plausible that KCs harbor TRP channels that exert a pivotal role in regulating intracellular calcium levels and ER homeostasis. Research has indicated that cannabinol (CBD) acts as a TRPV1 agonist, triggering calcium influx, augmenting ROS generation, and inducing cell death in breast cancer cell lines. In contrast, co-treatment with TRPV1 antagonists has been shown to enhance cell viability [120]. Furthermore, certain TRP channels, such as TRPV6, TRPC1, and TRPML1, may ameliorate ERS, while others, such as TRPV1, TRPC6, and TRPC3, may exacerbate ERS to promote cell death [121]. Ca^2+^ influx induced by KC P2X7 receptor activation plays multiple roles in liver inflammation [122]. However, whether TRP channels, which also mediate the influx of cations (especially Ca^2+^), play an important role in the activation of KCs deserves further exploration.

### 3.4. TRP Channels in Endothelial Cells

The liver contains two distinct microvascular structures, namely the portal vein vessels and the hepatic sinusoid. The former is characterized by a complete vascular structure comprising continuous vascular endothelial cells arranged in the lumen of a basement membrane. In contrast, the latter is composed of liver sinusoidal endothelial cells (LSECs) that lack a basement membrane and possess open windows or cross-cell pores, thereby forming permeable barriers that facilitate direct communication with hepatocytes. This unique feature enables LSECs to absorb oxygen, micronutrients, and macronutrients from the blood [123].

To date, there exists no documentation regarding the manifestation of TRP channels within LSECs. Given the fenestrae and sieve plates present in LSECs, ions within the bloodstream can traverse in and out of these cells with ease. Nevertheless, the vascular structures akin to the portal vein, hepatic artery, and hepatic vein hold significant importance in the liver. Studies indicate that portal vein angiogenesis impedes liver fibrosis [124]. Simultaneously, the TRP channel superfamily members that are expressed in the LSECs constituting these vessels in the liver exhibit comparable expression patterns and functions to those expressed in vascular endothelial cells in other organs. Consequently, we shall elucidate the expression and function of TRP channels in endothelial cells apart from LSECs.

It has been reported that ECs express the majority of identified mammalian TRP isomers, including TRPC1, 3, 4, 5, 6, and 7; TRPV1, 2, and 4; TRPP1 and 2; TRPA1; and TRPM1, 2, 3, 4, 6, 7, and 8, at both gene and protein levels [125,126]. The activation of TRP channels by vascular endothelial growth factor (VEGF) has been demonstrated to increase EC calcium concentration, thereby regulating the signaling pathway that leads to angiogenesis. The capacity of endothelial progenitor cells (EPCs) to self-renew, proliferate, and differentiate into ECs is closely associated with angiogenesis. The discernible reduction in the functional activity of EPCs derived from TRPC1-knockout mice suggests that TRPC1 plays a crucial role in the process of angiogenesis [62]. The vascular endothelial growth factor receptor in human umbilical vein ECs is responsive to VEGF stimulation. This receptor mediates the activation of TRPC3 via its downstream DAG, which subsequently activates the Na^+^/Ca^2+^ exchanger in a reverse mode, resulting in Na^+^ influx and ultimately promoting angiogenesis [64]. Furthermore, the inhibition of TRPC4 and TRPC6 significantly impedes the VEGF-induced cationic current and generation of capillaries in vitro [65,70]. In human microvascular ECs, TRPV1 participates in simvastatin-induced Ca^2+^ influx, activates the CaMKII signal, and enhances the formation of the TRPV1-enos complex, leading to NO production and in vitro angiogenesis. TRPV4 has also been shown to regulate angiogenesis by stimulating endothelial cell proliferation and migration [60]. VEGF has been shown to stimulate the migration of ECs and induce ROS-dependent Ca^2+^ entry through the activation of TRPM2. In vivo studies have demonstrated that inhibiting TRPM4 can enhance tubular formation in the matrix and improve capillary integrity. In addition, silencing TRPM7 has been found to mimic the effects of Mg^2+^ deficiency on the growth and migration of microvascular ECs, indicating that magnesium and TRPM7 play a role in regulating angiogenesis [106,127]. Taken together, TRP channels are likely to play a significant role in the formation of intrahepatic blood vessels and contribute to the development and progression of liver fibrosis, liver tumors, and other related diseases.

## 4. Effects of TRP Channels on the Occurrence and Progression of Liver Diseases

### 4.1. TRP Channels in Liver Injury

Numerous acute and chronic liver injuries elicit liver inflammation by means of KC activation and recruitment of monocytes and neutrophils, which is facilitated by chemokines. Studies indicate that the hepatic concentrations of chemokines (Ccl2 and Cxcl2) and proinflammatory cytokines (TNF-α, il-1α, il-1β, and il-6) are markedly reduced in TRPV1 knockout mice and that neutrophil infiltration is attenuated [49]. Furthermore, there is speculation regarding the potential of the TRP channels to augment the inflammatory response subsequent to hepatocyte injury by regulating the biological activities of KCs, monocytes, and neutrophils, which includes cytokine and chemokine production, chemokine response, adhesion, and migration from the bloodstream to the damaged liver [44]. Acetaminophen (APAP) overdose represents a prevalent etiology of acute liver failure in developed nations, frequently necessitating liver transplantation for patient survival. Current studies have shown that TRP channel family members play an important role in APAP-mediated acute liver injury.

Cytochrome P450, located in the ER, catalyzes the conversion of APAP to N-acetyl-para benzoquinone (pBQ) imine (NAPQI). In instances where the accumulation of NAPQI surpasses the capacity for rapid neutralization via glutathione (GSH) coupling, a cascade reaction of liver cell necrosis is triggered through the interaction between NAPQI and multiple downstream proteins [128]. On the one hand, NAPQI has the ability to directly stimulate the TRPV4 channel and bind to free sulfhydryl groups located in cysteine residues of the TRPC1 and TRPV1 channels, thereby enhancing their oxidative modification and ultimately resulting in channel opening. On the other hand, NAPQI protein adducts are capable of regulating the function of the respiratory chain, generating peroxides (such as H_2_O_2_), which induce mitochondrial oxidative stress, and stimulating the activation of TRPM2 and TRPM7 channels by ROS production [45]. Research has indicated that TRPM2 can be transported to the PM of hepatocytes through lysosomal exocytosis in response to oxidative stress induced by H_2_O_2_ or APAP. The primary activator of TRPM2 is intracellular ADP ribose (ADPR), which binds to the TRPM2 NUDT9-h motif located at the C-terminus, leading to the opening of the channel pore [89,91,129]. TRPM2 is activated by H_2_O_2_ and other oxidants by augmenting the production of ADPR. Simultaneously, the production of intracellular ADPR responds to DNA damage caused by ROS by activating poly ADP ribose (pADPR) polymerase (PARP) [71]. The activation of multiple TRP channels expressed by hepatocytes leads to an increase in intracellular Ca^2+^ concentration, which further aggravates the mitochondrial oxidative stress reaction and finally leads to the activation of RIP3 and translocation of Drp1 and Bax to mitochondria. Bax-induced outer membrane permeability and Drp1-induced mitochondrial fission mediate the mitochondrial membrane permeability transition (MPT). Apoptosis-inducing factor (AIF) and endonuclease are released from the mitochondria and transferred to the nucleus, where they initiate nuclear DNA fragmentation [128] (Figure 1). In addition, the harmful effects of TRPM2 activation may be caused by not only Ca^2+^ entry but also TRPM2-mediated inflow of Na^+^ and outflow of K^+^. The accumulation of intracellular Na^+^ and the loss of K^+^ lead to the loss of PM potential and the activation of Na^+^-K^+^-ATPase, leading to further depletion of cell ATP levels and promoting cell necrosis [130].

N-acetylcysteine (NAC) is the only effective drug therapy for patients with APAP-induced liver injury, and it is considered to be able to supplement and replace the GSH required to bind NAPQI. The GSH inducer DMF attenuates the decrease in GSH content, increase in ROS levels, Ca^2+^ overload, and cell death-related events induced by APAP or H_2_O_2_. Thus, it may be used as a treatment method to replace NAC [45]. In addition, because APAP-mediated acute liver injury depends on Ca^2+^ influx mediated by multiple members of the TRP channel superfamily, it is suggested that targeted inhibition of these TRP channels may be a new therapeutic strategy for acute liver injury.

### 4.2. TRP Channels in Liver Fibrosis

Chronic and persistent liver injury resulting from viral infection, excessive alcohol consumption, metabolic etiology, and autoimmune liver disease typically leads to liver fibrosis, a condition marked by the activation of HSCs and excessive accumulation of ECM [131,132]. During the initial inflammatory stage, fibrosis serves as a beneficial mechanism for wound repair. This process is facilitated by the coordinated activation of coagulation, immune response, and fibroblast activation, which promote the regeneration of damaged tissue or the formation of fibrous scar tissue. However, prolonged inflammation can impede the synthesis, deposition, and degradation of ECM components, leading to fibrosis inhibition. Tissue injury and fibrogenic mediators, such as TGF-β and platelet-derived growth factor (PDGF), mediate the differentiation of stationary HSCs into proliferating myofibroblast-like cells. The activation of HSCs is accompanied by cellular changes, including the loss of vitamin A droplets, de novo expression of α-SMA, and overproduction and deposition of ECM components, such as type I collagen [133]. The TRP channels play an important role in the activation of HSCs and the development of liver fibrosis (Table 2).

TRPM7 shows significant permeability toward both Mg^2+^ and Ca^2+^ and is a crucial component in maintaining cellular Mg^2+^ homeostasis. Its expression in HSCs is associated with cell survival. Studies have demonstrated that the inhibition of TRPM7 channels through the use of Gd^3+^ and 2-aminoethoxydiphenyl borate (2-APB) results in increased apoptosis and reduced expression of α-SMA and Col1α1 in HSC-T6 cells [76,104]. The downregulation of TRPM7 mediates the inhibition of HSC-T6 cell proliferation and is associated with decreased expression of cyclin D1, cyclin-dependent kinase 4, and proliferating cell nuclear antigen [75]. HSCs are responsible for ECM secretion, deposition, and accumulation, and are a crucial contributor to liver fibrosis. Consequently, blocking TRPM7 to impede HSC activation and proliferation and induce apoptosis may represent a promising approach to prevent or reverse liver fibrosis.

Liu et al. used CCl4 and BDL to develop a mouse model of liver fibrosis. Their investigation revealed a notable increase in TRPM8 expression within liver fibrosis tissue. Furthermore, they observed a significant reduction in liver injury and fibrosis in TRPM8^−/−^ mice when compared with the control mice [46]. Subsequent investigation has revealed that TRPM8 knockdown results in reduced expression of S100A9, a factor associated with inflammation, and HNF4α, a regulator of gene expression specific to the liver. S100A9 is a constituent of the S100 protein family, which can be discharged from cells undergoing inflammation or damage and binds to toll-like receptor 4 or advanced glycation end product receptor, thereby stimulating an inflammatory reaction. Further, HNF4α is a central transcriptional regulator of hepatocyte gene expression, differentiation, and function maintenance and has been proven to play a central regulatory role in alleviating liver fibrosis [134]. The inhibition of TRPM8 has the potential to mitigate liver fibrosis by downregulating the expression of the pro-inflammatory factor S100A9 and promoting the expression of HNF4α. Moreover, the inhibition of TRPM8 has been observed to decrease the number of F4/80-positive cells and suppress the activation of HSCs and cholangiocytes by reducing gene expression levels of IL-6, IL-1β, TNFα, and MCP-1 in a mouse liver fibrosis model. Cholangiocytes, which proliferate in response to endogenous and exogenous stimuli, actively participate in intrahepatic inflammation and repair processes, thereby promoting the progression of liver fibrosis [46,135,136] (Figure 2).

In a previous in vitro experiment, knockout of TRPV3 with siRNA hindered DNA synthesis and HSC proliferation and increased cell apoptosis [57]. Further, activation of TRPV3 has been reported to upregulate the inflammation-related gene *Olr1*, resulting in increased expression of the protein LOX-1. LOX-1 facilitates the uptake of oxidized low-density lipoprotein (ox-LDL) into endothelial cells and reduces ox-LDL content in the blood. In vivo investigations have demonstrated a positive correlation between LOX-1 expression and inflammatory factor levels [56,137]. Following TRPV3 activation, a conspicuous infiltration of macrophages has been observed in the area of fibrotic lesions. Macrophage polarization can be induced by various injury factors, and the release of inflammatory factors and chemokines by activated macrophages can exacerbate liver inflammation and fibrosis [138]. Consequently, a reduction in TRPV3 expression or functional level may ameliorate the inflammatory response and mitigate the proliferation of fibrotic tissue.

In vivo, the activation of the TRPV4 channel has been observed to exacerbate liver fibrosis, whereas the inhibition of the TRPV4 channel has been shown to alleviate liver fibrosis. Notably, the TRPV4 channel has been found to be significantly upregulated in the tissues of patients with liver fibrosis and CCl4-treated rats. In addition, under the influence of TGF-β1, TRPV4 expression in HSC-T6 cells has been found to increase, potentially because of direct regulation by miR-203 [107]. TRPV4 knockdown using siRNA has been demonstrated to strongly inhibit the proliferation of activated HSC-T6 cells, as downregulation of TRPV4 appears to inhibit the autophagy of TGF-β1-treated HSC-T6 cells [59]. Autophagy is a crucial mechanism for cellular proliferation and apoptosis and also serves as a catalyst for HSC activation. Suppression of autophagy can impede HSC proliferation and facilitate HSC apoptosis. Upon stimulation with TGF-β1, cultured HSC-T6 cells have previously exhibited significant activation of the AKT signaling pathway. Studies indicate that the AKT signaling pathway is intricately linked to autophagy, and inhibition of the AKT signaling pathway activation can promote HSC apoptosis [139]. Thus, TRPV4 may inhibit HSC apoptosis by regulating the activation of the autophagy-dependent AKT signaling pathway, and targeting TRPV4 may become an effective treatment strategy to prevent the progression of liver fibrosis.

The HSC cell line lx-2 cells exhibit a significant upregulation of TRPC6, which is dependent on the activation of NICD under hypoxic conditions. Hypoxia, being an environmental stressor, triggers the activation of oxygen-sensitive HSCs and facilitates the onset of liver fibrosis. The activation of the fiber formation process under hypoxic conditions is mediated by the hypoxia-inducible factor (HIF), which has been identified as a key mediator. Further, the upregulation of hypoxia-inducible factor 1α (HIF1α) has been reported to result in an increase in the nuclear localization of NICD, which subsequently induces the expression of TRPC6. The influx of Ca^2+^ through TRPC6 channels directly activates the Ca^2+^-sensitive protein phosphatase calcineurin, which in turn triggers the synthesis of ECM proteins via its downstream transcriptional effector, activated T cell nuclear factor (NFAT) [69]. Previous research has demonstrated that Smad3 plays a significant role in the fibrotic response of HSCs, and it is believed that TRPC6 promotes the expression of α-SMA and collagen through the activation of SMAD2/3 [140] (Figure 3).

To conclude, the upregulation of TRPM7, TRPM8, TRPV3, TRPV4, and TRPC6 is significantly associated with the onset and progression of liver fibrosis. Targeting these TRP channels in clinical settings may offer a potential strategy for the prevention or reversal of liver fibrosis, thereby safeguarding the liver against the deleterious effects of advanced cirrhosis and HCC.

### 4.3. TRP Channels in Liver Cancer

The predominant forms of liver cancer are HCC and intrahepatic cholangiocarcinoma, with HCC comprising approximately 90% of all liver cancer cases. Other types of liver cancer include mixed hepatocellular cholangiocarcinoma, fibrolamellar HCC, pediatric hepatoblastoma, and metastatic liver cancer [141]. HCC is the sixth most prevalent cancer type and the third leading cause of cancer-related death. Because of the liver’s robust compensatory capacity, HCC is frequently detected in intermediate and advanced stages, resulting in suboptimal surgical resection opportunities and diminished therapeutic outcomes. Furthermore, the persistent high recurrence rate of HCC poses a formidable challenge, rendering it to be a significant global health concern [142,143]. Hence, there is a pressing need to identify novel biological markers and therapeutic targets for the timely detection and improved prognosis of HCC. Given the prevalence of HCC in liver cancer and the focus of this investigation, this article solely presents an overview of the expression and function of TRP channels in HCC.

Recent research suggests that cancer may be classified as a channel disease, as tumor cell survival, death, and movement are regulated through ion channels and transporters. Of particular significance is the role of Ca^2+^ as a crucial messenger in the regulation of cell proliferation, apoptosis, transcription, migration, and angiogenesis. Studies have demonstrated that the disruption of intracellular Ca^2+^ homeostasis creates a unique microenvironment within the liver, facilitating the activation of various signaling pathways, including Wnt/bcatenin, TP53/cell cycle, telomere maintenance, chromatin regulatory factors, and others. These pathways drive mutagenesis, leading to the accelerated proliferation of hepatocytes [144,145,146]. TRP channels play a significant role in regulating various cellular physiological and pathophysiological processes in cancer, primarily by modulating the expression levels of functional TRP proteins to dynamically regulate intracellular ion concentrations, rather than through TRP gene mutations. Notably, multiple TRP channels have been identified in HCC at both the gene and protein levels. Specifically, both Huh7 and HepG2 human hepatoma cell lines express mRNAs for TRPC1, TRPC6, TRPV1, TRPV2, TRPV4, TRPM4, TRPM6, TRPM7, and TRPM8, whereas Huh7 alone expresses TRPV3 and TRPM5 [85].

#### 4.3.1. TRPC

TRPC1 is co-located with lipid raft proteins, such as caveolin-1, and is considered to be an important component of SOCE. SOCE can maintain the Ca^2+^ levels in the ER. When SOCE is inhibited, it lacks storage and supplementation, and cell growth and proliferation are inhibited [147]. Cancer cell migration has been reported to be induced by TGF-β through the stimulation of intracellular Ca^2+^ release and Ca^2+^ entry via TRPC1 and Na^+^/Ca^2+^ exchangers. Further, TRPC1 silencing resulted in the inhibition of Huh7 cell proliferation, while both Ca^2+^ release and SOCE in the ER were upregulated, suggesting that TRPC1 exerts a regulatory effect on SOCE. The overexpression of TRPC1 in HCC is linked to unfavorable prognoses of afflicted patients. Bioinformatics analysis suggests that TRPC1 impedes retinol metabolism and other vital metabolic processes by amplifying shared signaling pathways that facilitate tumor proliferation and the expression of genes such as ABI2, MAPRE1, YEATS2, MTA3, and TMEM237 [148]. Following TRPC1 silencing, there was a significant decrease in the expression of cell cycle-regulating genes CDK11A/11B and URGCP, accompanied by a notable increase in the expression of survival-promoting genes ERBB3 and FGFR4 [149,150]. Similarly, TRPC1 inhibition through shRNA or SKF 96365 in D54MG glioma cells resulted in a reduction in cell proliferation. In vivo experiments have further demonstrated that the inhibition of TRPC1 expression led to a reduction in the size of lateral abdominal tumors [151].

TRPC5 exhibits high expression levels in paracancerous tissues, and its activation is potentially mediated by regulation of the Akt/IκB/NF-κB signaling pathway, which inhibits macrophage differentiation and causes an increase in M1 macrophage infiltration. The release of proinflammatory cytokines and chemokines (such as TNF-α, IL-12, and iNOS) by M1 macrophages results in the inhibition of cell proliferation, reduction in tumor cell invasiveness, and suppression of malignant phenotypes, which ultimately leads to the promotion of favorable prognoses for patients with HCC [66,67,68].

The present study reveals that the expression of TRPC6 mRNA and protein is significantly upregulated in liver cancer tissues compared with normal liver tissues. Furthermore, in vitro experiments using HepG2 and Huh7 cells demonstrate that TGF-β plays a crucial role in the development of HCC by promoting the formation of the TRPC6-NCX1 complex, which leads to an elevation in intracellular Ca^2+^ levels and subsequently enhances the migratory and invasive properties of HCC cells [47]. Overexpression of TRPC6 in HCC cells results in the continuous accumulation of intracellular free calcium, which in turn stimulates epithelial–mesenchymal transition (EMT), Hif1-α signal transduction, and DNA damage repair. This process enables HCC cells to acquire multi-drug resistance (MDR), which may be mediated by the calcium-dependent TRPC6/calcium/STAT3 pathway. The use of siRNA or SKF 96365 to inhibit TRPC6 can alleviate MDR induced by various stimuli. Further, targeted inhibition of TRPC6 in combination with adriamycin in vivo demonstrates a synergistic anti-tumor effect [152]. Furthermore, significant inhibition of HCC cell proliferation can be achieved by blocking TRPC6. This suggests that TRPC6 targeting can potentially serve as a novel anti-tumor approach by alleviating MDR and suppressing HCC proliferation.

#### 4.3.2. TRPV

Research has demonstrated that the activation of TRPV1 by capsaicin is responsible for the elevation in intracellular Ca^2+^ levels, production of ROS, and activation of the STAT3 pathway in HepG2 cells. This presents a promising avenue for targeting and eliminating HCC cells as well as regulating their migration [153]. MMP1 upregulation in HCC patients is noteworthy because of its association with poor prognosis. However, the activation of TRPV1 can increase MMP1 expression, which contradicts previous studies linking TRPV1 overexpression to longer disease-free survival in liver cancer patients [48,58,154,155]. Currently, no study has established a direct correlation between TRPV1 expression and MMP1 upregulation in HCC cells, necessitating further research to elucidate the underlying mechanism by which TRPV1 regulates HCC cells. 

During the progression from hepatitis to cirrhosis to liver cancer, there is a gradual increase in both TRPV2 mRNA and protein levels. In contrast, poorly differentiated tumors exhibit reduced expression of TRPV2 mRNA and protein compared with well-differentiated HCC tumors. In addition, a correlation exists between TRPV2 expression and portal vein invasion, thereby suggesting that TRPV2 serves as a potent prognostic marker for patients with HCC [52]. ROS facilitates the upregulation of TRPV2 mRNA and protein levels in HepG2 and Huh7 cells, resulting in the suppression of survival-promoting factors, including Akt and Nrf2, and the stimulation of death-promoting factors, namely, p38 and JNK1, during the initial phase of apoptosis [53]. Furthermore, the expression of TRPV2 has been found to be associated with the desiccation of HCC cells. Notably, the downregulation of TRPV2 has been observed to significantly enhance the colony-forming capacity of HepG2 cells and the expression of stem cell markers CD133 and CD44 in hepatoma cell lines. Conversely, the overexpression of TRPV2 has been shown to diminish the formation of globules and colonies [54]. Hagit et al. presented a novel approach to enhance the clinical application of doxorubicin, a chemotherapeutic drug, in the liver cancer cell line BNL1 ME. They revealed that the co-administration of cannabinol (CBD) or 2-APB with doxorubicin resulted in significant augmentation of doxorubicin accumulation in BNL1 ME cells [55]. The limited penetration of doxorubicin through the cell membrane due to its weak alkaline chemical nature led to a concentration gradient, with the concentration of doxorubicin outside the cell being approximately three times higher than that inside the cell [156]. The clinical utility of doxorubicin is constrained by its irreversible cardiac toxicity. The BNL1 ME cell line, derived from mice with HCC, exhibits the expression of the macroporous cation channel receptor TRPV2. The agonists of TRPV2, namely, CBD and 2-APB, facilitate the activation of TRPV2, thereby facilitating the entry and accumulation of doxorubicin within BNL1 ME cells. In addition, CBD has been demonstrated to inhibit P-gp ATPase, thereby reducing the efflux of doxorubicin from cells [157]. Furthermore, the absence of noteworthy TRPV2 protein expression in cardiomyocytes or hepatocytes implies that localized hepatic administration of this drug combination will exclusively affect HCC cells without any off-target consequences. However, the non-specific TRPV2 agonists, 2-APB and CBD, may trigger other ion channels, thereby disturbing the ion homeostasis in normal cells [158] (Figure 4).

In vitro, the TRPV4 channel of HCC cells was inhibited by the specific antagonist HC067047, resulting in the inhibition of cell proliferation and promotion of cell apoptosis. This effect was achieved with the weakening of EMT and inactivation of p-ERK. The pro-apoptotic effect of TRPV4 was found to be blocked by influencing the transcription and translation of apoptosis-related genes Bax and Bcl2, ultimately leading to the activation of caspase 3 [159]. Similarly, Lee et al. reported that TRPV4 inhibitors can effectively inhibit the migration and invasion of 4T07 human breast cancer cells with high expression of TRPV4 [160].

Venom peptide (Tv1) treatment in HCC cell line 1’s MEP cells resulted in a significant reduction in COX-2 and prostaglandin E2 (PGE2) levels, which was found to be associated with TRPV6/TRPC6-mediated calcium-dependent apoptosis [40]. The dephosphorylation of calcineurin activates the Ca^2+^-dependent transcription factor NFAT, which regulates COX-2 expression in various cancer cells [161,162]. Overexpression of COX-2 results in elevated levels of PGE2, which subsequently binds to the EP receptor. The activation of the EP receptor in turn is responsible for facilitating angiogenesis, inhibiting apoptosis, stimulating cell growth, and enhancing the potential for invasion and metastasis of tumor tissue [163]. TV1 has the potential to impede the influx of Ca^2+^ into neoplastic cells via TRPV6 by interacting with extracellular recruitment sites comprising E518, E519, and N548 residues in the TRPV6 tetramer and via selective filters located in the D542 side chain of the pore region. This mechanism may effectively hinder the aforementioned process of tumor initiation and advancement [40] (Figure 5).

The combination of drugs targeting the COX-2/PGE2 axis and traditional anti-tumor drugs presents a promising approach for liver cancer treatment. Specifically, the combination of selective COX-2 inhibitor meloxicam and T7 polypeptide has a potent anti-tumor effect on liver cancer in mice [164]. Nevertheless, the multifaceted physiological and pathological functions of PGE2 necessitate caution in the use of COX-2 inhibitors, as long-term intake may result in adverse effects on the kidney and cardiovascular system [165,166]. Hence, there is an urgent need to devise novel targets. Despite the incomplete elucidation of the TRPV6 mechanism that hinders apoptosis of hepatoma cells, it presents fresh perspectives for investigators. The potential for TRP channel-targeting drugs to supplement selective COX-2 inhibitors in cancer therapy necessitates additional exploration.

#### 4.3.3. TRPM

Functional TRPM7-like channels have been identified in the proliferating and polarized rat hepatoma cell line WIF-B, and their regulation is dependent on the concentration of cytoplasmic Ca^2+^ through CaMKII-dependent processes. These channels may contribute to the survival of WIF-B cells and may also be involved in the migration of hepatoma cells. In particular, TRPM7-mediated Ca^2+^ influx has been shown to enhance the activity of calpain, which is known to play a role in the migration of HepG2 cells [167] (Table 2).

**Table 2 ijms-24-12509-t002:** The role of TRP channels in liver-related diseases.

Liver Diseases	TRP Channels	Function	Ref.
Hepatic injury	TRPV1, TRPV4, TRPC1, TRPM2, TRPM7	Enhance the inflammatory response after hepatocyte injury	[44,45,89,91]
Hepatic fibrosis	TRPM7, TRPM8, TRPV3, TRPV4, TRPC6	Promote the activation and proliferation of HSCs and aggravate liver fibrosis	[46,57,59,76,104,140,168]
HCC	TRPC1, TRPC6, TRPV6, TRPV4, TRPM7	Promote the progression of HCC	[40,47,148,153,159,167,169]
	TRPC5, TRPV1, TRPV2	Reduce the aggressiveness of tumor cells	[52,55,66,67,68]

Abbreviations: HCC, hepatocellular carcinoma; HSCs, hepatic stellate cells.

### 4.4. Drugs for TRP Channels

Numerous drugs function by targeting TRP channels during the treatment of diseases. As TRP is extensively expressed in the sensory neurons, brain, and skin, earlier studied drugs were formulated to alleviate chronic pain. For instance, the TRPV1 agonist capsaicin (8-methyl-N-vanillyl-6-nonenamide), possessing analgesic and anesthetic properties, is frequently used to mitigate post-therapeutic neuralgia, diabetic neuralgia, rheumatoid arthritis pain, and osteoarthritis pain. Nevertheless, a drawback of TRPV1 agonists is their potential to elicit a robust initial pain response, thereby constraining their clinical application. Consequently, non-stimulatory TRPV1 agonists, including N-(3-methoxy-4-hydroxybenzyl) oleamide (NE19550) and MDR-652, have been investigated by researchers to circumvent these untoward effects. However, these compounds are currently undergoing animal experimentation and require further clinical validation [170]. An ultrapotent capsaicin analog, resiniferatoxin, has demonstrated efficacy in alleviating pain in canines afflicted with osteosarcoma, as well as in human subjects diagnosed with metastatic cervical cancer, and is presently undergoing clinical trials. Researchers aim to achieve sustained pain relief for individuals suffering from severe osteoarthritis and chronic cancer [17]. In the context of TRPV1 receptor-targeted drugs, a therapeutic dose of a TRPV1 agonist functions as an antagonist of the TRPV1 receptor. In particular, a low dose of capsaicin stimulates the release of inflammatory mediators via TRPV1, whereas a therapeutic dose inhibits such a release. Consequently, the initial generation of TRPV1 antagonists was associated with febrile reactions and burns [18]. However, the second-generation TRPV1 antagonist JNJ-39439335 (mavatrep) has demonstrated significant efficacy in reducing pain during exercise in patients with knee arthritis in clinical trials [171]. Eucalyptol (1,8-cineol), another TRPV1 antagonist, has demonstrated pain relief in mice with gouty arthritis, whereas the TRPV1 antagonist SB-366791 (N-(3-methoxyphenyl)-4-chlorocinn amide) exhibited pain relief in rats with tooth pain. In addition to TRPV1-targeting drugs, TRPA1 antagonists, such as A-967079, have shown the potential to treat neuropathic pain in animal experiments. Furthermore, GRC17536 (Glenmark Pharmaceuticals), a TRPA1 antagonist that is currently undergoing clinical trials, has been used to alleviate diabetic neuropathic pain. Other investigations have indicated that small-molecule TRPM8 antagonists may alleviate cold-induced pain [17].

TRPV1 antagonists, such as SB-705498 (N-(2-bromophenyl)-N’-[((R)-1-(5-trifluoromethyl-2-pyridyl) pyrrolidin-3-yl)]urea), have the potential to mitigate capsaicin-induced cough in addition to pain management. The reversal of acetylcholine-induced airway hyperreactivity in mice with Ovalbumin-induced allergic asthma with the TRPA1 antagonist HC-030031 indicates the therapeutic potential of TRPA1 in the inhibition of asthma. Further, the oral TRPA1 antagonist GDC-0334 has demonstrated efficacy in suppressing cough, airway hyperresponsiveness, and edema in clinical trials. In contrast, menthol, an over-the-counter cough suppressant, acts as a TRPM8 agonist and may effectively inhibit citric acid-induced cough by activating TRPM8 [17]. The TRPV1 agonists, capsaicin and resiniferatoxin, have been found to have potential applications beyond pain management. In particular, they may be used to desensitize individuals with a neurogenic bladder, resulting in increased bladder capacity and reduced instances of incontinence [172]. In addition, topical resiniferatoxin desensitization may be used for the prevention of premature ejaculation in males [173]. The present study investigated the efficacy of TRPV1 antagonists, namely, GRC-6211, and TRPM8 antagonists, namely, RQ-00434739 and KRP-2529, in mitigating hyperactivity in a chronically inflamed bladder. In addition, the TRPV4 agonist, GSK1016790A, was examined for its potential to improve underactive bladders, and TRPV4 antagonists were evaluated for their ability to ameliorate overactive bladders [174]. TRP drugs have demonstrated potential for application in dermatology. Notably, the TRPV1 antagonist PAC-14028, currently in phase III clinical trials, has been found to improve the skin barrier function and alleviate pruritus in patients with atopic dermatitis. However, topical administration of this drug may increase the risk of infection in patients. Nonetheless, the TRPM8 agonist menthoxypropanediol cream has been shown to alleviate pruritus in humans, while the topical administration of the TRPA1 antagonist HC-030031 has been found to alleviate UV-induced burn injury in mice. Furthermore, TRPV4 inhibition has also been found to alleviate skin pruritus. TRP-targeted drugs exhibit potential in treating ocular and central nervous system diseases. In animal experiments, the TRPV4 antagonist HC-067047 was effective in treating ocular matrix opacities following alkali burn [175]. Similarly, the TRPM2 antagonist JNJ-28583113 has shown promising results in improving cognitive dysfunction in mice after cerebral ischemia [17]. 

TRP-targeting drugs have demonstrated the ability to enhance glucose metabolism. Specifically, in murine models of type 2 diabetes, the small molecule antagonist N-(4-Tertiarybutylphenyl)-4-(3-cholorphyridin-2-yl) tetrahydropyrazine-1(2H)-carbox-amide (BCTC) targeting TRPV1 has been found to improve insulin secretion in response to glucose stimulation. In addition, clinical trials have shown that the TRPV1 antagonist XEN-D0501, when used in conjunction with metformin, exhibits favorable glucose-lowering effects [176]. The efficacy of numerous drugs in cancer treatment is linked to TRP channels. For instance, Vacquinol-1, an immunosuppressant, triggers cell death in glioblastoma cells via its TRPM7 ATP-inducible inhibitory effect. Another example is gemcitabine, a cytotoxic drug that exhibits heightened cytotoxicity when combined with anti-TRPM7 siRNA, which may be triggered through the p16 (CDKN2A) and WRN mRNA pathway [151].

Animal studies have validated the association between TRP channels and organ fibrosis. In particular, the administration of TRPC6 antagonist BI 749327 has been shown to effectively impede the progression of kidney and myocardial fibrosis. This is achieved by suppressing the activation of the nuclear factor of activated T cells (NFAT), thereby hindering the expression of pro-hypertrophic genes [177]. Despite the extensive application of TRP-targeting drugs in animal testing and clinical trials, there remains a paucity of drugs used for liver diseases. Current research efforts are concentrated on treating liver fibrosis. In mouse experiments, the TRPM8 antagonist M8-B hydrochloride exhibited potential in inhibiting the activation of cholangiocytes by downregulating the expression of inflammatory factor S100A9 and upregulating the expression of HNF4α, which ultimately led to the amelioration of hepatobiliary inflammation and liver fibrosis [46]. Furthermore, investigations conducted on a CCl4-induced liver fibrosis mouse model demonstrated that the TRPV4 agonist GSK1016790A exacerbated liver fibrosis, whereas the TRPV4 antagonist HC-067047 effectively ameliorated liver collagen fiber deposition and the degree of liver lobular disorder [178]. Similarly, drofenine, a TRPV3 selective agonist, exacerbated liver fibrosis in a CCl4-induced mouse liver fibrosis model, while forsythoside B, a TRPV3 inhibitor, significantly mitigated liver fibrosis, although the underlying mechanism warrants further investigation [57]. TRP channels have potential as a drug target for the treatment of liver fibrosis and drug-induced hepatotoxic injury. The inhibition of TRPV4, for instance, has been shown to mitigate the hepatotoxic side effects resulting from excessive APAP in mice with acute liver failure [130].

## 5. Discussion

This study examines the expression and function of the TRP channel family in diverse liver cells and their implications for the onset and progression of liver-related ailments. The liver is susceptible to various pathogenic factors, and the influx of cations through TRP channels initiates or exacerbates a cascade of reactions causing hepatocyte damage. For instance, TRPV1 is involved in the generation and secretion of chemokines and proinflammatory cytokines in the liver and facilitates the infiltration of neutrophils, thereby intensifying the inflammatory response following liver injury. The pathogenesis of APAP-mediated acute liver injury involves the activation of TRPV1, TRPV4, TRPC1, TRPM2, and TRPM7 channels, which trigger the influx of Ca^2+^ into hepatocytes and subsequent cellular damage. In the context of liver fibrosis, the upregulation of TRPC6, TRPV3, TRPV4, and TRPM7 channels promotes the activation of HSCs and the production and accumulation of ECM components, including α-SMA and COL1A1. This mechanism significantly contributes to the progression of liver fibrosis. The activation and upregulation of TRPC1, TRPC6, TRPV4, TRPV6, and TRPM7 channels have been identified as contributing factors toward the proliferation and migration of HCC cells. In contrast, TRPC5, TRPV1, and TRPV2 channels have been found to impede macrophage differentiation, suppress HCC cell proliferation, and ultimately decrease tumor cell invasiveness through the Akt/IκB/NF-κB signaling pathway, STAT3 pathway, and Akt/Nrf2 signaling pathway, respectively. These channels have been associated with favorable patient outcomes.

The TRP superfamily exhibits restricted expression patterns, but its diverse tissue distribution renders it influential in the majority of cells, tissues, and organs in the human body. Recently, several TRP superfamily members have been the focus of targeted drug development for various diseases. Notably, TRPV1, which is prominently expressed in primary sensory neurons, plays a crucial role in mediating afferent pain stimuli and neurogenic inflammation. Following agonist stimulation, TRPV1 undergoes desensitization. Currently, TRPV1-expressing neurons exhibit a distinctive lack of response to repeated capsaicin stimulation and other extraneous stimuli [179], making it a singular drug target. Current clinical trials have used resiniferatoxin, a super capsaicin analog, as a “molecular scalpel” to alleviate severe osteoarthritis pain and chronic refractory pain in cancer patients. In addition, the TRPA1 antagonist exhibits therapeutic potential in patients with neuropathic pain. Notably, the TRPA1 antagonist GRC17536 demonstrated a significant reduction in the pain score among patients with painful diabetic polyneuropathy in the non-denervated group, without any adverse effects. However, the compound’s bioavailability/pharmacokinetics pose a challenge, leading to the completion of only phase II clinical trials [17]. The activation of TRPV4 has been demonstrated to result in persistent detrusor and bladder contractions, suggesting that the use of a TRPV4 agonist may be beneficial in the treatment of bladder hypotonia [174]. Moreover, TRPM4 has been identified as a critical factor in cerebral artery myogenic contraction. The application of siRNA to silence Trpm4 expression has been shown to decrease infarct volume in a rat model of permanent stroke, indicating that Trpm4 inhibition enhances the integrity of the blood–brain barrier following ischemic stroke reperfusion [127]. Modulation of TRP channel expression has been observed in various cancer types, including HCC, and is a potential therapeutic target. Notably, TRPV1 expression was found to be elevated in high-grade astrocytoma tissues compared with normal brain tissues. However, the administration of high doses of capsaicin was found to induce neuronal death via Ca^2+^ overload. Therefore, the use of TRPV1 agonists may hold promise in the eradication of this particular brain tumor [180].

Trpm5^−/−^ mice maintain normal weight with a carbohydrate-rich diet and consume less alcohol, indicating the importance of TRPM5 antagonists in obesity and/or alcohol use disorders [181]. In contrast, the TRPM8 agonist icilin increased energy consumption and decreased body weight in mice [182]. The liver is widely recognized as the primary organ responsible for lipid metabolism, and lipid metabolites may serve as crucial regulators of TRP channel function. In particular, lysophosphatidic acid (LPA) levels have been observed to increase following tissue injury, and this increase in LPA levels activates TRPV1 via direct interaction with the channel’s C-terminal. These findings suggest that LPA plays a pivotal role in the transduction of pain. The activation of TRPV2 by lysophosphatidylcholine (LPC) and lysophosphatidylinositol (LPI) has been shown to induce calcium influx and increase cell migration in the prostate cancer cell line PC3, indicating the potential pathological role of TRPV2 in prostate cancer [183]. In addition, farnesyl pyrophosphate (FPP) has been identified as an endogenous activator of TRPV3, which can elicit pain in vivo. Nitrogen-containing bisphosphonates have been demonstrated to inhibit FPP synthase, providing relief in certain types of bone cancers and neuropathic pain [184,185].

Alterations in TRP channel expression levels have been observed in liver-related pathologies, which have been shown to impact disease progression and prognosis. Nonetheless, the possibility that TRP channel regulation is a consequence of liver disease cannot be denied. Regrettably, no TRP agonist or antagonist has yet undergone clinical trials as a potential therapeutic intervention, particularly for liver diseases. The broad expression and diverse biological roles of TRP channels have impeded TRP channel drug development, primarily because of the occurrence of undesirable adverse reactions on the intended targets. Empirical evidence has demonstrated that achieving adequate specificity for clinically efficacious interventions while avoiding unacceptable adverse reactions is a challenging task. For instance, the first-generation TRPV1 antagonists AMG517 and MK2295 were withdrawn from clinical trials because of the emergence of fever reaction and burns, respectively [186,187]. Further, the phase II efficacy test conducted to evaluate the effectiveness of AZD1386 in treating osteoarthritis-related pain was terminated because of inadequate analgesic activity. Consequently, future research should be focused on developing specialized drug delivery systems, such as selective agonists or antagonists, to mitigate unacceptable adverse reactions [188]. Notably, nanocarriers have emerged as promising drug delivery systems for targeted liver treatment. In particular, DOX-loaded galactosylated chitosan nanoparticles have demonstrated potential as a selective drug formulation for treating HCC [189]. The Gal OSL nanocarrier has demonstrated efficacy in mitigating liver lipid accumulation, insulin resistance, and triglyceride (TG) accumulation through the regulation of the AMPK/SIRT/FAS/SREBP1c signaling pathway. The intervention of Gal OSL/Res has the potential to restore normal health status in mice with NAFLD and prevent the progression of the condition to severe liver disease [190].

## 6. Conclusions

Despite the difficulty in creating clinically effective TRP modulator drugs for liver-related illnesses, the substantial potential benefits of pursuing this avenue of research are evident. This is because of the pathogenic role of TRP channels in liver diseases, as well as the increasing number of individuals afflicted with such conditions annually.

## Figures and Tables

**Figure 1 ijms-24-12509-f001:**
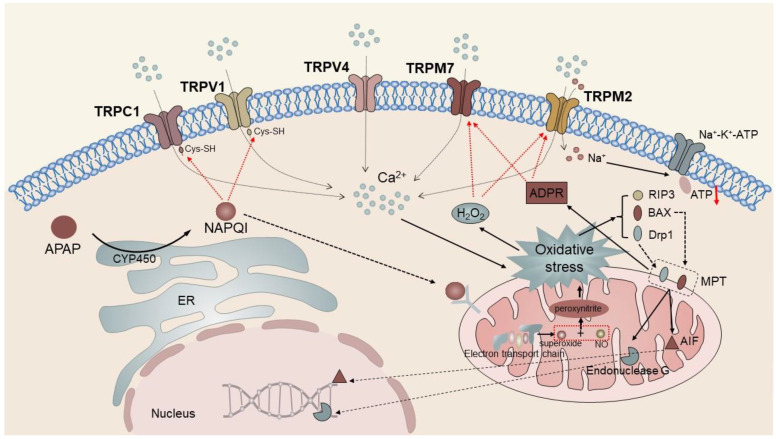
TRPC1, TRPV1, TRPV4, TRPM2, and TPM7 mediate APAP-induced liver injury. The intracellular metabolite of APAP, NAPQI, directly activates TRPC1 and TRPV1 to trigger Ca^2+^ influx and induce mitochondrial oxidative stress. It eventually leads to the activation of RIP3 and translocation of Drp1 and Bax to the mitochondria, triggering mitochondrial membrane permeability transition. In addition, NAPQI protein adducts modulate respiratory chain function similarly inducing mitochondrial oxidative stress responses. TRPM2 and TRPM7 channels activated by ROS and ADPR further aggravate intracellular cation concentrations, forming a vicious cycle and ultimately causing apoptosis.

**Figure 2 ijms-24-12509-f002:**
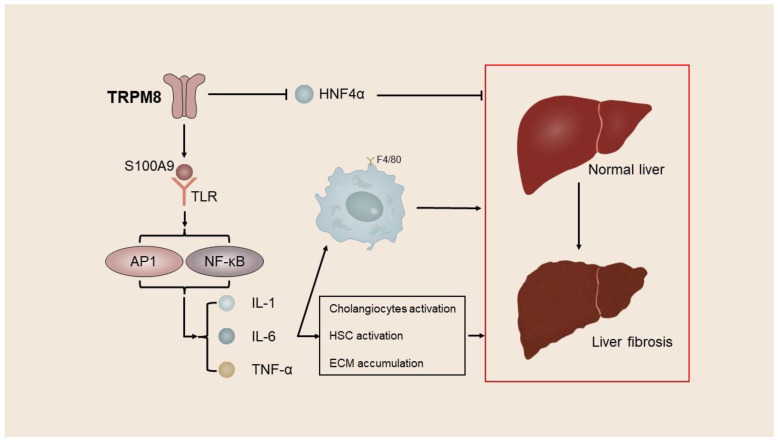
TRPM8 mediates the process of liver fibrosis. Activation of TRPM8 upregulates the inflammation-related factor S100A9 and downregulates the liver-specific gene expression regulator HNF4α while increasing the number of F4/80-positive cells, which in turn promotes the activation of HSCs and cholangiocytes and thereby mediates liver fibrosis.

**Figure 3 ijms-24-12509-f003:**
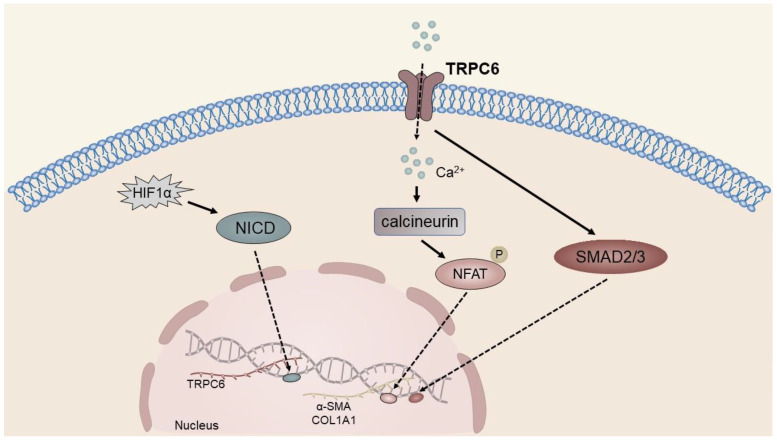
TRPC6 mediates the process of liver fibrosis. TRPC6 is significantly upregulated under hypoxic conditions in a manner dependent on NICD activation. TRPC6 promotes α-SMA and collagen expression through the activation of calcineurin and SMAD2/3, leading to liver fibrosis.

**Figure 4 ijms-24-12509-f004:**
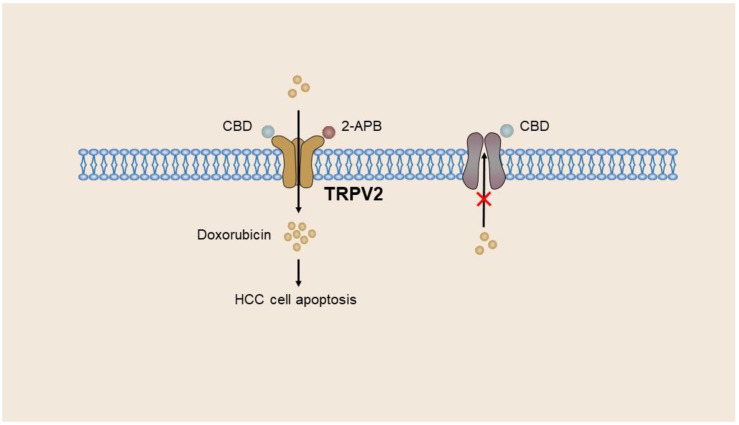
Adriamycin enters BNL1 ME cells of the hepatoma cell line through TRPV2. CBD and 2-APB act as agonists of TRPV2 to activate and open the TRPV2 channel, promoting the entry and massive accumulation of doxorubicin in BNL1 ME cells. Meanwhile, CBD inhibits P-gp ATPase, thereby reducing doxorubicin removal from cells.

**Figure 5 ijms-24-12509-f005:**
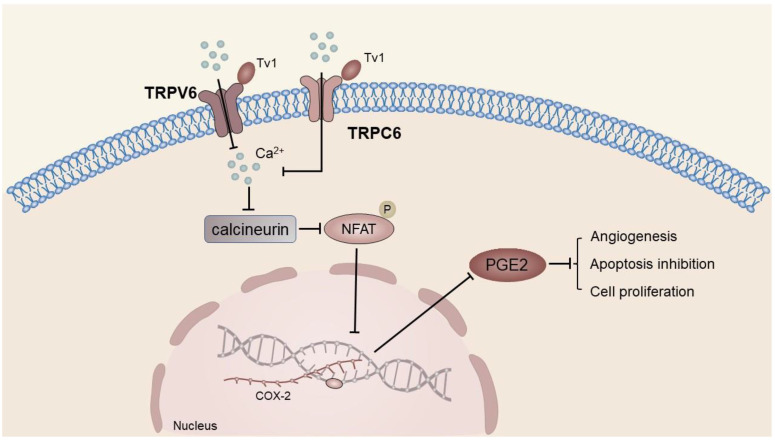
COX-2 and PGE2 levels were significantly decreased in venom peptide (Tv1)-treated 1MEA cells. Tv1 prevented Ca^2+^ influx after binding to TRPV6 and TRPC6, and the transcription factor NFAT could not be dephosphorylated by calcineurin, thus downregulating COX-2 expression.

**Table 1 ijms-24-12509-t001:** Expression of TRP channels and their role in liver-related diseases.

TRP Channel	Expression	Related Diseases	Main Functions	Ref.
**TRPV1**	Hepatocytes, HepG2 cells, ECs	hepatitis, hepatic injury, HCC	Regulates the stability of microtubule and cell migration; manipulates the biological function of KCs; mediates APAP-induced liver injury; and promotes angiogenesis.	[44,48,49,50,51]
**TRPV2**	HepG2 cells, Huh-7 cells, BNL1 ME cells	HCC	Mediates HCC cell survival and cell stemness; and mediates doxorubicin entry into BNL1 ME cells.	[52,53,54,55]
**TRPV3**	HSCs	hepatic fibrosis, cirrhosis	Regulates expression of inflammation-related gene Olr1 and proliferation of HSCs.	[56,57]
**TRPV4**	Hepatocytes, HepG2 cells, HSC-T6 cells, ECs	hepatic injury, hepatic fibrosis, HCC	Mediates APAP-induced liver injury; mediates activation and proliferation of HSCs; attenuates EMT and induceds inactivation of p-ERK; and promotes angiogenesis.	[45,58,59,60]
**TRPV6**	1MEA cells	HCC	Regulates the expression of COX-2.	[40]
**TRPC1**	H4-IIE cells, Huh-7 cells, ECs	hepatic injury, HCC	Regulates the volume of hepatocytes; mediates APAP-induced liver injury; regulates the SOCE and cellular activity of HCC cells; and promotes angiogenesis.	[45,61,62]
**TRPC3**	KCs, ECs	IRI	Mediates calcium overload and activation of KCs; and promotes angiogenesis.	[63,64]
**TRPC4**	ECs	-	Promotes angiogenesis.	[65]
**TRPC5**	Paracancerous tissues	HCC	Macrophage differentiation is inhibited by regulating the Akt/IκB/NF-κB signaling pathway.	[66,67,68]
**TRPC6**	lx-2 cells, HepG2 cells, Huh-7 cells, 1MEA cells, ECs	hepatic fibrosis, HCC	Mediates expression of α-SMA and COL1A1; mediates migration and MDR of HCC cells; regulates the expression of COX-2; and promotes angiogenesis.	[69,70]
**TRPM2**	Hepatocytes, ECs	hepatic injury	Mediates apoptosis and necrosis pathways; mediates APAP-induced liver injury; and mediates EC migration.	[45,71,72]
**TRPM7**	Hepatocytes, WIF-B cells, HSC-T6 cells	hepatic injury, hepatic fibrosis, HCC	Mediates hepatocyte proliferation and ROS-induced apoptosis; involved in TRAIL-induced HSC apoptosis; and mediates APAP-induced liver injury.	[45,73,74,75,76]
**TRPM8**	Hepatocytes	hepatic fibrosis	Downregulates the expression of S100A9 and promotes the expression of HNF4α.	[46]
**TRPML1**	Hepatocytes	-	Mediates lysosome release of Ca^2+^ to maintain intracellular calcium homeostasis.	[77]

Abbreviations: HCC, hepatocellular carcinoma; KCs, Kupffer cells; APAP, acetaminophen; HSCs, hepatic stellate cells; EMT, epithelial–mesenchymal transition; SOCE, store-operated calcium entry; IRI, ischemia-reperfusion injury; MDR, multi-drug resistance; ROS, reactive oxygen species; ECs, endothelial cells.

## Data Availability

The datasets used and/or analyzed during the current study are available from the corresponding author upon reasonable request.

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
