# Peer review of "Role of TRP Channels in Liver-Related Diseases"

_ijms, 2023, doi:10.3390/ijms241512509_

Round 1
Reviewer 1 Report
In this manuscript, Liu et al. summarized the function of TRP channels in liver diseases. This is an important and interesting field. The authors must carefully revise the manuscript, since there are many mistakes. In addition, it is hard to understand many sentence, due to the low English quality.
1. There are many careless misses. The authors must carefully correct the manuscript.
2. English quality is low. The manuscript must be edited by native speakers.
3. The authors should show the abbreviation of each word when they appear at the first time. Because the number of the abbreviations is too much, it is difficult to discriminate in total table.
4. Introduction section: It is difficult to understand the intention of the sentence 'While no targeted drugs aimed・・・・・・'
5. Connection between Introduction and section 2 is not clear.
6. Table 1 should include all the points in section3.
7. Section 4: The authors must summarize this section in table.
8. Discussion section is disorganized. The authors must re-write and show the clear conclusions.
Extensive correction is necessary.
Author Response
Reviewer 1:
In this manuscript, Liu et al. summarized the function of TRP channels in liver diseases. This is an important and interesting field. The authors must carefully revise the manuscript, since there are many mistakes. In addition, it is hard to understand many sentence, due to the low English quality.
- There are many careless misses. The authors must carefully correct the manuscript.
Answer:
We appreciate the reviewer’s suggestions. We have revised and improved the full text with the assistance of a native English speaker.
- English quality is low. The manuscript must be edited by native speakers.
Answer:
We apologize for the overlooks and are very grateful for your kind suggestions. We have revised and improved the manuscript text with the assistance of a native English speaker.
- The authors should show the abbreviation of each word when they appear at the first time. Because the number of the abbreviations is too much, it is difficult to discriminate in total table.
Answer:
We appreciate and agree with the reviewer’s comments. We have added definitions for all abbreviations at the first appearance in the text.
- Introduction section: It is difficult to understand the intention of the sentence 'While no targeted drugs aimed・・・・・・'
Answer:
We appreciate the reviewer’s valuable comments. We have revised the sentence to make it easier to understand. The corrections are made in lines 92 to 94.
- Connection between Introduction and section 2 is not clear.
Answer:
We are very grateful for your valuable suggestions. We have revised the text for clarity and better flow in the revised manuscript. The added text ranges from lines 107 to 110.
- Table 1 should include all the points in section3.
Answer:
We appreciate the reviewer’s suggestions. We have checked and added the section "TRP channels in endothelial cells" in Table 1.
- Section 4: The authors must summarize this section in table.
Answer:
We appreciate the reviewer’s suggestions. We have summarized and added Table 2 summarizing Section 4. The new table is placed between sections 4.3 and 4.4.
- Discussion section is disorganized. The authors must re-write and show the clear conclusions.
Answer:
We appreciate the reviewer’s valuable comments. We have revised the Discussion and added a Conclusion section. (We realized that the last paragraph of the Discussion section conveys the conclusion of the study, so we named it "Conclusion".) The correction begins on line 878.

Reviewer 2 Report
The presented review is a detailed analysis of the available data on the involvement of TRP channels in the pathogenesis of the liver. The authors presented detailed information on liver cells with an analysis of the expression of various TRP channels in them, discussed the relationship of individual subtypes of channels with the development of pathological conditions and possible molecular mechanisms underlying them. Additional information is provided on the therapeutic potential of TRP channel modulators and the possibility of their use for the relief of liver diseases. The review is very high quality, well written and enough easy to understand. There are no significant comments. Minor remarks below can be easily corrected.
1. section "Abbreviations" - 2APB decoding given twice
2. Incorrect channel decoding given - TRPC (typical), TRPV (vanillin), TRPM(melatadine), TRPA (anchor protein), TRPP (polycystic), TRPML (mucin), and TRPN (Drosophila NOMPC)
The accepted decryption is TRPC ("C" for canonical), TRPV ("V" for vanilloid), TRPM ("M" for melastatin), TRPA ("A" for ankyrin), TRPP ("P" for polycystic), TRPML ("ML" for mucolipin), TRPN ("N" for no mechanoreceptor potential C or NOMPC),
3. in a number of places the concept of “the TRP channel” occurs, if there are many members of this subfamily, it would be more correct to indicate the TRP channals or TRPs in the plural (pp. 7, 11, 12,19,22)
4. page 9 “The transient receptor potential vanillin 3 (TRPV3)”… must be corrected to “The transient receptor potential vanilloid 3 (TRPV3)”
5. as follows from the text, figures 2 and 3 should be interchanged, and also figures 4 and 5 should be interchanged.
6. The section "Discussion" is more logical to rename to "conclusion"
Author Response
Reviewer 2:
The presented review is a detailed analysis of the available data on the involvement of TRP channels in the pathogenesis of the liver. The authors presented detailed information on liver cells with an analysis of the expression of various TRP channels in them, discussed the relationship of individual subtypes of channels with the development of pathological conditions and possible molecular mechanisms underlying them. Additional information is provided on the therapeutic potential of TRP channel modulators and the possibility of their use for the relief of liver diseases. The review is very high quality, well written and enough easy to understand. There are no significant comments. Minor remarks below can be easily corrected.
1. section "Abbreviations" - 2APB decoding given twice
Answer:
Thank you for bringing this to our notice. We have removed the duplication in the revised manuscript. Changes are made on lines 39, 49, and 51.
- Incorrect channel decoding given - TRPC (typical), TRPV (vanillin), TRPM(melatadine), TRPA (anchor protein), TRPP (polycystic), TRPML (mucin), and TRPN (Drosophila NOMPC)
The accepted decryption is TRPC ("C" for canonical), TRPV ("V" for vanilloid), TRPM ("M" for melastatin), TRPA ("A" for ankyrin), TRPP ("P" for polycystic), TRPML ("ML" for mucolipin), TRPN ("N" for no mechanoreceptor potential C or NOMPC),
Answer:
We are very grateful for your valuable comment. We have corrected the wrong channel decoding in the revised manuscript. The corrections are made in lines 78 to 82.
- in a number of places the concept of “the TRP channel” occurs, if there are many members of this subfamily, it would be more correct to indicate the TRP channals or TRPs in the plural (pp. 7, 11, 12,19,22)
Answer:
We highly appreciate your kind suggestions. We have checked and corrected the term in the revised manuscript.
- page 9 “The transient receptor potential vanillin 3 (TRPV3)”… must be corrected to “The transient receptor potential vanilloid 3 (TRPV3)”
Answer:
We appreciate your kind suggestions. We have revised the terminology in the revised manuscript as per your comment (line 327).
as follows from the text, figures 2 and 3 should be interchanged, and also figures 4 and 5 should be interchanged.
Answer:
We appreciate the reviewer’s suggestions. We have interchanged the figures as requested.
The section "Discussion" is more logical to rename to "conclusion"
Answer:
We appreciate the reviewer’s valuable comments. We have revised the Discussion and added a Conclusion section. (We realized that the last paragraph of the Discussion section conveys the conclusion of the study, so we named it "Conclusion".) The correction begins on line 878.

Reviewer 3 Report
In the manuscript entitled “Role of TRP Channels in Liver-related Diseases”, the authors review the expression and roles of the TRP ion channels in liver cells and their involvement in liver-related disorders. Given the complexity involved, the author has produced many positive and welcome outcomes. The literature review offers a useful overview of current research and policy, and the resulting bibliography provides a very useful resource for current practitioners. This study well written and the illustrations are excellent and very didactic. The topic touched upon in the article is relevant. The scientific content of the manuscript justifies its publication, but some additions and modifications will significantly improve the quality of the article.
I point out specific areas in this manuscript that the authors can consider to improve this manuscript:
Introduction, pag.2:
- Please change “In recent times” with “Lately”
- Here and throughout, always leave a space between words or between words and a reference in the text. Please, check the whole text.
- They are further divided into seven families….”TRPC (typical)” should be TRPC “(canonical)”; “TRPM (melatadine)” should be “TRPM (melastatin)”.
- “TRP channels exhibit a broad distribution……as reported in the literature”. Here some references are needed. For instance: DOI: 10.1038/s41392-023-01464-x; DOI: 10.1002/cphy.c110026.
- “Moreover, TRPs facilitate the modulation of apoptosis …. in cancer and the immune system”. Here some references are needed. For instance: DOI: 10.1038/s41419-020-03256-5; doi: 10.1007/s10549-020-05673-8; doi: 10.1155/2019/7362875; doi: 10.3390/cells7070070.
- “While no targeted drugs aimed at TRPs have been employed in clinical trials, numerous compounds targeting TRPV1, TRPV4, TRPA1, and TRPM8 channels have been incorporated into clinical trials”. This sentence is not clear, please rephrase
Chapter 2, pag.3
- Here and throughout, please use the italic style for “in vivo” and “in vitro”
Chapter 3, pag.7
- urthermore,….should be Furthermore”
- “The present study demonstrates that TRPV1 activation-induced Ca2+ influx is involved in the migration of human hepatoblastoma HepG2 cells that have been pre-treated with hepatocyte growth factor (HGF) through calcium-sensitive targets, including myosin light chain kinase, gel protein, non-actin, or calcineurin”. This sentence is not clear, please rephrase.
Chapter 3, pag.8
- “Furthermore, the phosphorylation forms”…maybe you can say “phosphorylated”
Figures
- The order of the figures in the text is wrong. You should move figure 3 to the place of figure 2 and you should move figure 5 to the place of figure 4
The authors should check for few typos, there are also few paragraphs missing some words or having some extra words
Author Response
Reviewer 3:
In the manuscript entitled “Role of TRP Channels in Liver-related Diseases”, the authors
review the expression and roles of the TRP ion channels in liver cells and their involvement
in liver-related disorders. Given the complexity involved, the author has produced many
positive and welcome outcomes. The literature review offers a useful overview of current
research and policy, and the resulting bibliography provides a very useful resource for current
practitioners. This study well written and the illustrations are excellent and very didactic. The
topic touched upon in the article is relevant. The scientific content of the manuscript justifies
its publication, but some additions and modifications will significantly improve the quality
of the article.
I point out specific areas in this manuscript that the authors can consider to improve this
manuscript:
Introduction, pag.2:
- Please change “In recent times” with “Lately”
Answer:
We appreciate the reviewer’s suggestions. We have revised the text as per your suggestion in the manuscript (line 61).
- Here and throughout, always leave a space between words or between words and a reference
in the text. Please, check the whole text.
Answer:
We appreciate the reviewer’s suggestions. We have checked the full text and added spaces between words and references.
- They are further divided into seven families….”TRPC (typical)” should be TRPC
“(canonical)”; “TRPM (melatadine)” should be “TRPM (melastatin)”.
Answer:
We appreciate the reviewer’s valuable comments. We have corrected the channel name abbreviations and definitions (lines 78 to 82).
- “TRP channels exhibit a broad distribution……as reported in the literature”. Here some
references are needed. For instance: DOI: 10.1038/s41392-023-01464-x; DOI:
10.1002/cphy.c110026.
Answer:
We are very grateful for your valuable suggestions. We have added references as requested (line 85).
- “Moreover, TRPs facilitate the modulation of apoptosis …. in cancer and the immune
system”. Here some references are needed. For instance: DOI: 10.1038/s41419-020-03256-
5; doi: 10.1007/s10549-020-05673-8; doi: 10.1155/2019/7362875; doi:
10.3390/cells7070070.
Answer:
We appreciate the reviewer’s suggestions. We have added references as requested in the revised manuscript (line 90).
- “While no targeted drugs aimed at TRPs have been employed in clinical trials, numerous
compounds targeting TRPV1, TRPV4, TRPA1, and TRPM8 channels have been
incorporated into clinical trials”. This sentence is not clear, please rephrase
Answer:
We appreciate the reviewer’s suggestions. We have revised the sentence for clarity and better understanding (lines 92 to 94).
Chapter 2, pag.3
- Here and throughout, please use the italic style for “in vivo” and “in vitro”
Answer:
We appreciate the reviewer’s suggestions. We have checked the full text and used italics for "in vivo" and "in vitro".
Chapter 3, pag.7
- urthermore,….should be Furthermore”
Answer:
We appreciate the reviewer’s suggestions. We have corrected the word in the revised manuscript (line 213).
- “The present study demonstrates that TRPV1 activation-induced Ca2+ influx is involved in
the migration of human hepatoblastoma HepG2 cells that have been pre-treated with
hepatocyte growth factor (HGF) through calcium-sensitive targets, including myosin light
chain kinase, gel protein, non-actin, or calcineurin”. This sentence is not clear, please
rephrase.
Answer:
We are very grateful for your valuable suggestions. We have rephrased the sentence to make it clearer (lines 229–233).
Chapter 3, pag.8
- “Furthermore, the phosphorylation forms”…maybe you can say “phosphorylated”
Answer:
We appreciate the reviewer’s valuable comments. We have revised the word choice in the manuscript as per your suggestion.
Figures
- The order of the figures in the text is wrong. You should move figure 3 to the place of figure
2 and you should move figure 5 to the place of figure 4
Answer:
We appreciate the reviewer’s valuable comments. We have interchanged the figures as requested.
The authors should check for few typos, there are also few paragraphs missing some words
or having some extra words
Answer:
We appreciate the reviewer’s suggestions. We have revised and improved the full text with the assistance of a native English speaker.

Round 2
Reviewer 1 Report
The authors tackled all our points and substantially improved the manuscript.
Reviewer 3 Report
The authors have addressed all my comments/suggestions. I found their responses quite satisfactory and the revised version has been much improved. I now recommend the paper for publication in IJMS